# Dual specificity phosphatase 7 drives the formation of cardiac mesoderm in mouse embryonic stem cells

**Stanislava Sladeček, Katarzyna Anna Radaszkiewicz, Martina Bőhmová, Tomáš Gybeľ, Tomasz Witold Radaszkiewicz, Jiří Pacherník** *

Department of Experimental Biology, Faculty of Science, Masaryk University, Brno, Czech Republic

* jipa@sci.muni.cz

**Data Availability Statement:** All relevant data are within the manuscript and its Supporting Information files.

## Abstract

Dual specificity phosphatase 7 (DUSP7) is a protein belonging to a broad group of phosphatases that can dephosphorylate phosphoserine/phosphothreonine as well as phosphotyrosine residues within the same substrate. DUSP7 has been linked to the negative regulation of mitogen activated protein kinases (MAPK), and in particular to the regulation of extracellular signal-regulated kinases 1 and 2 (ERK1/2). MAPKs play an important role in embryonic development, where their duration, magnitude, and spatiotemporal activity must be strictly controlled by other proteins, among others by DUSPs. In this study, we focused on the effect of DUSP7 depletion on the in vitro differentiation of mouse embryonic stem (ES) cells. We showed that even though DUSP7 knock-out ES cells do retain some of their basic characteristics, when it comes to differentiation, they preferentially differentiate towards neural cells, while the formation of early cardiac mesoderm is repressed. Therefore, our data indicate that DUSP7 is necessary for the correct formation of neuroectoderm and cardiac mesoderm during the in vitro differentiation of ES cells.

## 1. Introduction

The mitogen activated protein kinase (MAPK) pathway is one of the better described and vigorously studied signaling pathways. MAPK plays an important role in many cellular processes like proliferation, differentiation or apoptosis, and its function has been described in the contexts of animal development, cancer biology, immune response, to name just a few [1–4]. The MAPK family includes three kinase subfamilies—extracellular signal-regulated kinases (ERK), c-Jun N-terminal kinases (JNK), and p38 [5]. MAPKs are active when phosphorylated on both threonine and tyrosine residues within their activation loop (TxY motif) and can be inactivated by a number of phosphatases, among which are dual specificity phosphatases (DUSP) [6, 7].

The human genome encodes more than twenty members of the DUSP family, although their classification can sometimes differ in literature. DUSPs, similarly to MAPKs, have been studied in many types of cancer lines as well as in embryonic stem cells and animal

**Funding:** This research was supported by the Faculty of Science of Masaryk University (MUNI/A/ 1145/2017) and by the Czech Science Foundation (Project 18-18235S); SS was supported by a grant from the Czech Science Foundation (Project 19-16861S). https://www.muni.cz/en/about-us/ organizational-structure/faculty-of-science https:// gacr.cz/en/ The funders had no role in study design, data collection and analysis, decision to publish, or preparation of the manuscript.

**Competing interests:** The authors have declared that no competing interests exist.

development. They have been described as important for pluripotency [8–10], neural development [11], cardiac development [12], immunity [13], and cancer prognosis [14] etc.

Dual specificity phosphatase 7 (DUSP7) is a cytoplasmic phosphatase, which dephosphorylates extracellular signal-regulated kinases 1 and 2 (ERK1/2) [15]. The expression of DUSP7 is upregulated, either due to an increase in its expression or stability in some pathological conditions such as leukemia or breast cancer [16–20], where it is associated with poor prognosis. Although DUSP7 was studied in cancer cell lines and even sparked interest as a potential cancer drug target [21], it remains one of the less studied DUSPs.

In this study, we focused on the role of DUSP7 in mouse embryonic stem (ES) cells and its effect on cell differentiation in vitro. We observed that the depletion of DUSP7 did not change some of the basic characteristics of mouse embryonic stem cells. However, DUSP7 deficiency did lead to changes in cell differentiation through the formation of embryoid bodies. Specifically, differentiating DUSP7 KO cells expressed lower levels of markers typical for mesoderm and, later on, cardiomyocytes, and higher levels of markers typical for ectoderm and, later on, neural progenitors. Together, these data indicate that DUSP7 plays an important role in early neural and cardiac mesoderm development.

## 2. Material and methods

### 2.1. Cell culture and differentiation

The mouse ES cell line R1 was adapted to feeder-free culture. R1 ES and all genetically modified cell lines derived from them were cultivated as described previously [22]. Undifferentiated cells were cultivated in DMEM media supplemented with 15% FBS, 100 U/ml penicillin, 0.1 mg/ml streptomycin, and 1x non-essential amino acid (all from Gibco-Invitrogen) and 0.05 mM β-mercaptoethanol (Sigma), supplemented with 1 000 U/ml of leukemia inhibitory factor (LIF, Chemicon). Differentiation of the cells was induced by seeding them onto a non-adhesive surface in bacteriological dishes to form floating embryoid bodies or by seeding them in the form of hanging drops (400 cells per 0.03 ml drop), all in medium without LIF. After five days of the cultivation of embryoid bodies, they were transferred to adherent tissue culture dishes and cultivated in DMEM-F12 (1:1) supplemented with insulin, transferrin, selenium (ITS, Gibco-Invitrogene), and antibiotics (as above), referred to as ITS medium. Cells were cultivated for up to 21 days depending on individual experiments and the medium was changed every two days. In the case of experiments with genetically selected cardiomyocytes, on day 14 medium was supplemented with 0.5mg/ml of antibiotic G418. An ES cell line for genetically selected cardiomyocytes was prepared subsequently: R1_MHC-neor/pGK-hygro ES cells (referred to as HG8 cells), carrying the Myh6 promoter regulating the expression of neomycin phosphotransferase, were prepared by the transfection of R1 cells by MHC- neor/pGK-hygro plasmid (kindly provided by Dr. Loren J. Field, Krannert Institute of Cardiology, Indianapolis, US) [23].

### 2.2. Creating KO lines using CRISPR-Cas9

DUSP7-null mouse embryonic cell lines were prepared using the CRISPR-Cas9 system as previously described [24, 25]. Guide RNA was designed using the online CHOPCHOP tool [26]. Plasmid pSpCas9(BB)-2A-Puro (PX459) V2.0 (#62988, Addgene) with puromycin resistance was used as the target plasmid to carry the guide RNA. Plasmids were prepared as described previously [27]. For transfection, 24h after passaging, cells were transfected in a serum-free medium using polyethyleneimine (PEI) in a stoichiometry of 6 μl of PEI per 1 μg of DNA for 8h. Then, the medium was changed for medium with puromycin (Invivogen, 10ug/ml). Selection lasted for 24h, after which the medium was changed for fresh medium without

puromycin, and when formed, colonies of potentially KO cells were picked. Acquired KO cell lines were tested by PCR using the primers TGTTGTGTGAGTCCTGACCG and AGAGG-TAGGGCAGGATCTGG (337bp product) for the amplification of genomic DNA, and Hpy166II restriction enzyme (R0616S, New England BioLabs) (240bp and 97bp products), then verified by next generation sequencing using the Illumina platform, as described previously [28].

## 2.3. Cell growth

Cells were seeded at a concentration of 1000 cells/cm$^2$ and cultivated in full medium for up to 5 days. From day 3, cells were stained with crystal violet, as previously described [29]. After the colonies had dried, 10% acetic acid was added to the wells and incubated with shaking for 30min. The absorbance of the obtained solution was then measured at 550nm on a Sunrise Tecan spectrophotometer.

For proliferation analysis using the WST-8 assay, cells were seeded on a culture-treated flat bottom 96-well plate at concentrations of 1000, 500, 250, 125 and 67 cells/well and cultivated in full medium for 3 days. Cells were incubated with WST-8/Methoxy-PMS (MedChem Expres HY-D0831 and HY-D0937, final concentration 0,25mg/ml and 2,5ug/ml respectively) for 5 hours and absorbance was measured at 650 nm and 450 nm on Multiscan GO (Thermo Scientific).

For proliferation analysis using the EdU assay, cells were seeded at a concentration of 5000cells/cm$^2$ and cultivated in full medium for 3 days. Cell proliferation was measured using the Click-iT™ Plus EdU Alexa Fluor™ 488 Flow Cytometry Assay Kit (Thermo Fisher, C10632). Cells were treated with 10 μM EdU (5-ethynyl-2′-deoxyuridine) for one hour prior to harvesting and processed according to the kit manufacturer's instructions. The untreated cells of each analyzed line were used as a control. Cells were analyzed using Cytek® Northern Lights spectrum flow cytometry. 20,000 events were acquired per each sample the percentage of EdU positive cells was analyzed using SpectroFlo software (Cytek). Single cells were identified and gated by pulse-code processing of the area and the width of the signal. Cell debris was excluded by using the forward scatter threshold.

## 2.4. Analysis of gene expression by qRT-PCR

Total RNA was extracted by RNeasy Mini Kit (Qiagen). Complementary DNA was synthesized according to the manufacturer's instructions for RevertAid Reverse Transcriptase (200 U/μL) (EP0442, Thermo Fisher). qRT-PCR was performed in a Roche Light-cycler using the protocols for SyberGreen (Roche) or TaqMan (Roche). The protocol for primers using SyberGreen was as follows: an initial activation step at 95˚C for 5 min, followed by 40 cycles at 95˚C for 10 s, an annealing temperature (Table 1) for 10 s, and a temperature of 72˚C for 10 s, followed by melting curve genotyping and cooling. The protocol for primers using TaqMan was as follows: an initial activation step at 95˚C for 10 min followed by 45 cycles of 95˚C for 10 s, 60˚C for 30 s, and 72˚C for 1 s with data acquisition. Primer sequences, annealing temperatures, and the probes used are listed in Table 1. The gene expression of each sample was expressed in terms of the threshold cycle normalized to the average of at least two so-called house-keeping genes. These were *Actb* and *Tbp* in the case of the SybrGreen protocol, and *Rpl13a* and *Hprt* in the case of the TaqMan protocol.

## 2.5. Counting of cardiomyocytes

The relative number of cardiomyocytes in differentiating ES cell cultures was determined. For these experiments, cells that were initially differentiated in the form of hanging drops were used. After 5 days of differentiation, embryoid bodies were transferred to ITS medium in

**Table 1. Probes and sequences of primers and temperature used in quantitative RT-PCR.**

| Gene of interest | Forward primer 5′ → 3′ | Reverse primer 5′ → 3′ | UPL probe no | Ta (°C) |
|---|---|---|---|---|
| *Hprt* | tcctcctcagaccgcttttt | cctggttcatcatcgctaatc | #95 | 60 |
| *Rpl13a* | catgaggtcgggtggaagta | gcctgtttccgtaacctcaa | #25 | 60 |
| *Nkx2.5* | gacgtagcctggtgtctcg | gtgtggaatccgtcgaaagt | #53 | 60 |
| *Myh6* | cgcatcaaggagctcacc | cctgcagccgcattaagt | #6 | 60 |
| *Myh7* | cgcatcaaggagctcacc | ctgcagccgcagtaggtt | #6 | 60 |
| *Mef2c* | accccaatcttctgccact | gatctccgcccatcagac | #6 | 60 |
| *Gata4* | ggaagacaccccaatctcg | catggccccacaattgac | #13 | 60 |
| *T* | actggtctagcctcggagtg | ttgctcacagaccagagactg | #27 | 60 |
| *Mesp1* | acccatcgttcctgtacgc | gcatgtcgctgctgaagag | #89 | 60 |
| *Sox1* | ccagcctccagagcccgact | ggcatcgcctcgctgggttt | | 61 |
| *Actb* | gatcaagatcattgctcctcct | taaaacgcagctcagtaacag | | 60 |
| *Tbp* | accgtgaatcttggctgtaaac | gcagcaaatcgcttgggatta | | 60 |
| *Oct4* | agaggatcaccttgggggtaca | cgaagcgacagatggtggtc | | 61 |
| *Nanog* | aggacaggtttcagaagcaga | ccattgctagtcttcaaccactg | | 60 |
| *Zfp42* | gcacacagaagaaagcagga | cactgatccgcaaacacct | | 59 |
| *Fgf5* | aagtagcgcgacgttttcttc | ctggaaactgctatgttccgag | | 61 |
| *Klf4* | gactaaccgttggcgtgag | gggttagcgagttcgaaagg | | 60 |
| *Dusp7* | gcccatccgctccatcattccc | cagccgtcgtctcgcagcttc | | 62 |
| *Pax6* | cgggaaagactagcagccaa | gtgaaggaggagacaggtgtg | | 62 |
| *Afp* | tggttacacgaggaaagccc | aatgtcggccattccctcac | | 60 |
| *Gata1* | gaagcgaatgattgtcagca | cagcagaggtccaggaaaag | | 61 |
| *Gata2* | gggagtgtgtcaactgtggt | gcctgttaacattgtgcagc | | 61 |
| *Mash1* | ttctccggtctcgtcctactc | ccagttggtaaagtccagcag | | 62 |
| *Tubb3* | tgaggcctcctctcacaagta | gtcgggcctgaataggtgtc | | 62 |
| *Dusp6* | acctggaaggtggcttcagt | tccgttgcactattggggtc | | 62 |

24-well plates and cultivated for a total of 20 days. Before analyses, cells were washed with phosphate buffered saline (PBS), incubated in a 0.3% solution of Collagenase II (Gibco) in DMEM media without serum for 20 minutes, and then incubated in trypsin (0.25% in PBS-EDTA, Gibco) for 5 minutes. Trypsin was inactivated by adding DMEM media with FBS, and cells in this medium were transferred to a new 24-well plate and cultivated for a further 24h. Cells were then washed with PBS, fixed for 20min with 4% formaldehyde, permeabilized by 0.1% TWEEN 20 solution in PBS, and stained using anti-cardiomyocyte heavy myosin antibody (anti-MHC, clone MF20, kindly provided by Dr. Donald Fischman, Developmental Studies Hybridoma Bank, Iowa City, IA, USA). They were then visualized using anti-mouse IgG conjugate Alexa568 (Invitrogen). Nuclei were counterstained with DAPI (1 mg/l). Images were acquired using an Olympus IX-51 inverted fluorescence microscope (Olympus) or Leica TCS SP8 (Leica) confocal microscope. In each repetition at least five images were taken from at least two wells for each line and the ratio between red and blue signals was analyzed using ImageJ software. The analysis of whole embryoid body staining was performed on cells cultivated in the same manner, but cells were seeded on day 5 on cover slips and on day 20 cells were not disassociated, but the whole embryoid body was fixed and stained as described above. Representative images were acquired using Leica TCS SP8 (Leica) confocal microscope.

The relative number of cardiomyocytes was also determined using R1_MHC-neor/pGK-hygro ES cells (HG8 cells) and their DUSP7 KO clones (MHC-neo/DUSP7 KO; analysis of frame shift mutations in both *Dusp7* alleles in these cell lines by NGS shown Fig 1B), as

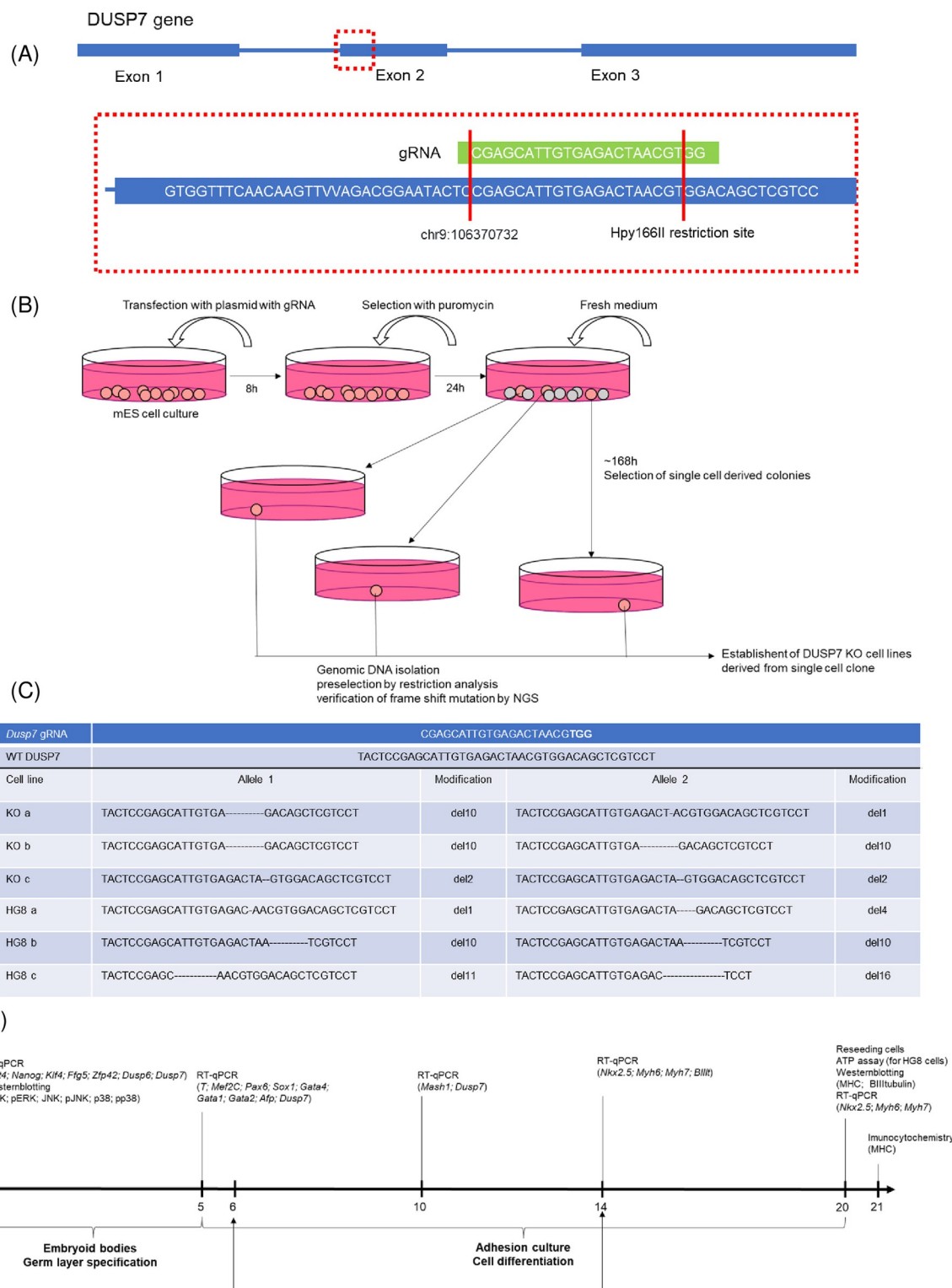

**Fig 1.** **(A)** Schematic representation of the guide RNA (gRNA) and *Dusp7* targeted area by the CRISPR-Cas9 system. **(B)** Schematic representation how the mouse embryonic KO cell lines were created and screened. **(C)** The guide RNA (gRNA) design and the sequences of verified deletions in the DUSP7 KO ES cell line edited by the CRISPR-Cas9 system are shown. **(D)** Schematic representation of the different experiments conducted in this study–at what time points different methods were employed and which markers were analyzed.

described previously [22]. Estimation of the relative number of viable cells after antibiotic selection was performed by ATP quantification in whole cell lysates. Cells were lysed in Somatic cell ATP releasing reagent for ATP determination (FLSAR-1VL, Sigma-Aldrich). Each cell lysate was mixed with Cellular ATP Kit HTS (155–050, BioThema) in the ratio of 1:1 and luminescence was analyzed using Microlite™ 1+ strips (Thermo Scientific) and Chameleon V (Hindex).

## 2.6. Small Interfering RNA (siRNA) transfection

Cells were transfected by commercially available siRNA DUSP6 (sc-39001) and DUSP7 (sc-61428) to knock down gene expression, or by related non-silencing control (all Santa Cruz Biotechnology, USA) using Lipofectamine RNAiMAX Reagents (Thermo Fisher Scientific Inc., USA) according to the manufacturer's instructions. Cells were harvested 24h after transfection and the expressions of selected proteins and posttranslational modification were analyzed by western blot [30].

## 2.7. Western-blot analysis

Cells were directly lysed in Laemmli buffer (100 mM Tris/HCl (pH 6.8), 20% glycerol, 1% SDS, 0.01% bromophenol blue, and 1% 2-mercaptoethanol). Western blotting was performed according to the manufacturer's instructions with minor modifications (SDS-PAGE run at 110 V, transfer onto PVDF membrane for 1 h at 110 V (BIO-RAD)). Membranes were blocked in 5% non-fat dry milk solution in TBS-T for 30 min and subsequently incubated overnight at 4˚C with primary antibodies listed in Table 2 (dilution 1:1000). Next, membranes were washed in TBS-T and incubated with HRP-conjugated secondary antibodies (Sigma-Aldrich). Immunoreactive bands were detected using ECL detection reagent kit (Merck-Millipore) and the FusionSL chemiluminescence documentation system (Vilber-Lourmat). Results were quantified by the densitometric analysis of Western blot bands using the Fiji distribution of ImageJ.

## 2.8. Isolation of mouse hearts

CD1 mice were maintained and bred under standard conditions and were used in accordance with European Community Guidelines on accepted principles for the use of experimental animals. Mouse hearts were isolated according to an experimental protocol that was approved by the National and Institutional Ethics Committee (protocol MSMT-18110/2017-5). Individual heart samples were prepared as described previously [22].

## 2.9. Statistical analysis

Data analysis was performed by GraphPad Prism. Data are expressed as mean ± standard deviation (SD). Statistical analysis was assessed by t-test and by one- or two-way ANOVA, and by Bonferroni's Multiple Comparison post hoc test. Values of $P < 0.05$ were considered to be statistically significant ($^*$ p $< 0.05$).

## 3. Results

### 3.1. Absence of DUSP7 does not affect the phenotype of ES cells

In order to determine the effect of DUSP7 on ES cells, we created DUSP7 knock out (KO) cell lines using the CRISPR-Cas9 system, as previously described [27, 31] (see experimental design on Fig 1A and 1B). These mutated cell lines were verified by NGS (Fig 1C) and were then used for all subsequent experiments (see experimental set up in Fig 1D). All used DUSP7 KO cell

**Table 2. Primary antibodies used for western-blot analysis.**

| Antibody | Catalog number | Company | Description | pub. identifier |
|---|---|---|---|---|
| p-ERK1/2 | CS-4370S | Cell Signaling Technology | molyclonal, rabbit | AB_2315112 |
| ERK1/2 | CS-4695S | Cell Signaling Technology | monoclonal, rabbit | AB_390779 |
| PARP | 9532 | Cell Signaling Technology | monoclonal, rabbit | AB_659884 |
| JNK | sc-571 | Santa Cruz Biotechnology | polyclonal, rabbit | AB_632385 |
| pJNK | sc-6254 | Santa Cruz Biotechnology | monoclonal, mouse | AB_628232 |
| p-38 | 9212 | Cell Signaling Technology | polyclonal, rabbit | AB_330713 |
| pp-38 | 9211 | Cell Signaling Technology | rabbit | AB_331641 |
| MHC | anti-MHC, clone MF20 | Developmental Studies Hybridoma Bank | monoclonal, mouse | AB_2147781 |
| βIIItubulin | Ab7751 | Abcam | monoclonal, mouse | AB_306045 |
| DUSP6 | sc-377070 | Santa Cruz Biotechnology | monoclonal, mouse | AB_2802089 |
| DUSP6 | 3058 | Cell Signaling Technology | polyclonal, rabbit | AB_2246226 |
| Vinculin | V9264 | Sigma | monoclonal, mouse | AB_10603627 |
| β-Actin | sc-47778 | Santa Cruz Biotechnology | monoclonal, mouse | AB_2714189 |

lines had either the same or different mutations in individual alleles, but in both cases the result was a frame shift mutation (Fig 1C).

We were able to cultivate all obtained DUSP7 KO cell lines in vitro for a substantial time (40+ passages), during which we did not observe any morphological difference between KO and wild type (WT) control (CTR) cell lines. To determine whether DUSP7 KO cell lines proliferate at a similar rate we stained them on three consecutive days using crystal violet, the results indicating that there were no significant differences in growth rate between WT and KO cell lines, nor between individual KO lines (Fig 2A). The same proliferation was confirmed by the EdU assay and the WST-8 assay (Fig 2B and 2C).

Next, we determined whether DUSP7 KO cells retain their stem cell phenotype by testing whether they differ from WT cells in expressing markers which are known to change if pluripotency is compromised–specifically, *Oct4*, *Nanog*, *Klf4*, *Zfp42* and *Fgf5* [32–36]. Cells were kept in standard culture conditions for ES cells for 5–40 passages before these markers were analyzed. We did not observe any statistical differences in the expressions of the given markers between any of these lines (Fig 2D). On the basis of the above, we conclude that the depletion of DUSP7 does not affect the proliferation rate of ES cells nor their pluripotent phenotype.

## 3.2. DUSP7 regulates germ layer specification in differentiating ES cells

To further test their stem cell-like properties, we studied the ability of DUSP7 KO cells to differentiate. All cell lines were able to form embryoid bodies (EBs) of the same shape and size in hanging drops or in cell suspension culture (Fig 3A). In 5-day-old EBs, transcripts of all three germ layers were determined–namely, *Sox1* and *Pax6* as markers for ectoderm/ neuroectoderm; *T*, *Mesp1*, *Mef2c*, *Gata4*, *Gata1* and *Gata2* as markers for mesoderm; and *Afp* as a marker for entoderm [37–42]. In DUSP7 KO cells, the expressions of *T*, *Mesp1* and *Gata4* were decreased compared to WT cells, while the expressions of *Sox1* and *Pax6* were increased. Excluding the expression of *Afp* in one DUSP7 KO cell line, we did not observe significant differences in the levels of *Mef2c*, *Gata1*, *Gata2*, or *Afp* between KO and WT cells. The increase in *Afp* was observed in only one of the DUSP7 KO cell lines, indicating that it might be an artefact typical only for this individual line (Fig 3B). These data show that DUSP7 is required for the correct formation of ectoderm and mesoderm during in vitro differentiation of ES cells.

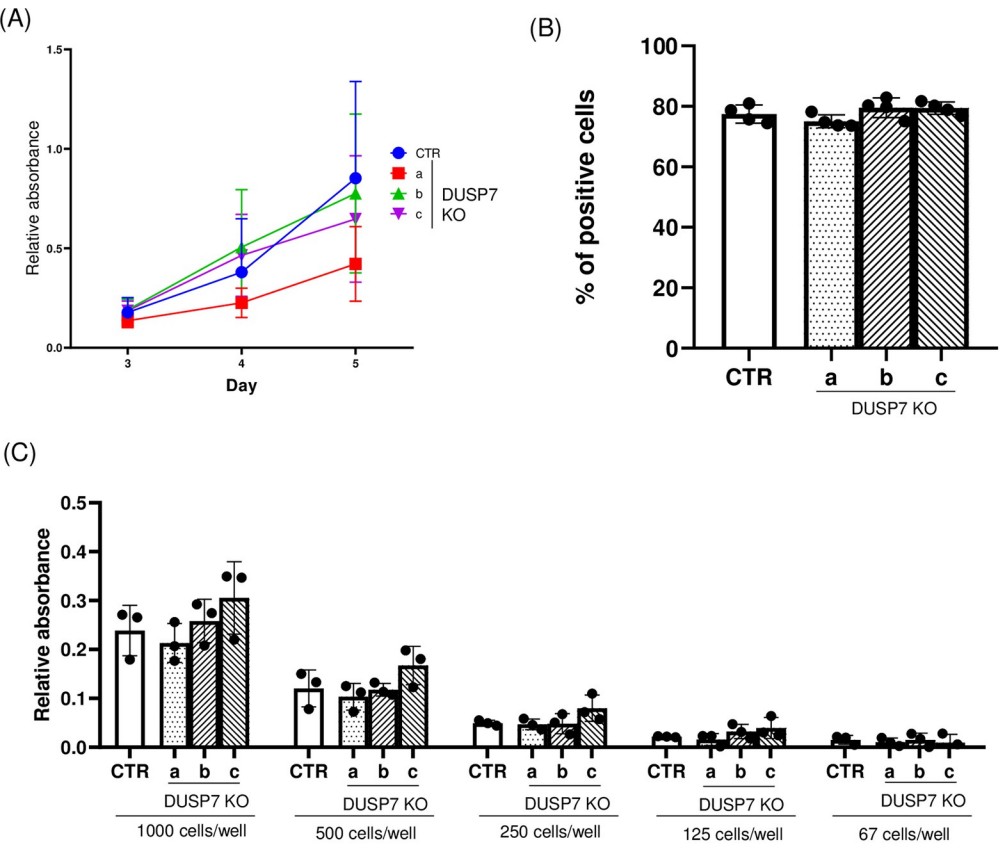

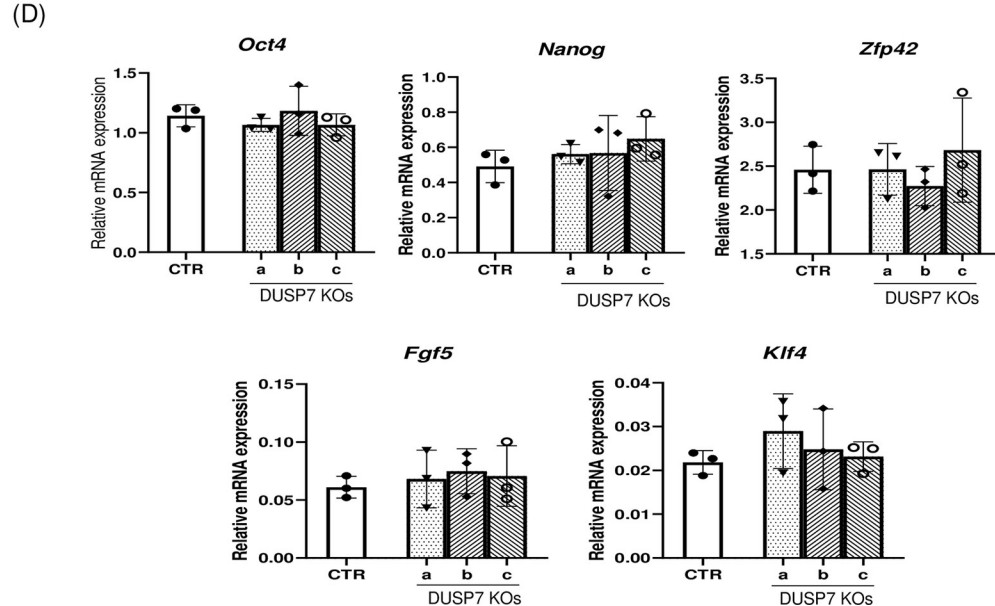

**Fig 2. DUSP7 KO mouse embryonic stem cells retain the basic characteristic of the cell line. (A)** To assess the growth curve of wild type and DUSP7 KO cell lines, 1000 cells/cm$^2$ were seeded on a fresh plate on day 0 and plates were stained with crystal violet staining on days 3,4 and 5 to assess their growth curve. The experiment was repeated three times and the value for individual experiments represents the average value obtained from four plates. **(B)** To assess proliferation rate, wild type and DUSP7 KO cell lines were seeded at a concentration 5000 cells/cm$^2$ and after 3 days of cultivation they

were treated with EdU for 1h. Graph represents mean ± SD of four independent replicates. (C) To test whether the proliferation rate of DUSP7 KO cells will be different compared to wild type cells, based on cell density, cells were seeded in a 96 well plate at a concentration of 1000, 500, 250, 125 and 67 cells/well. Cells were cultivated for 3 days after which they were incubated with WST-8/Methoxy-PMS for 5 hours and relative absorbance was measured. Graph represents mean ± SD of three independent replicates. **(D)** To check whether the DUSP7 KO cell lines retained their pluripotency, known markers of pluripotency *Oct4*, *Nanog*, *Zfp42*, *Ffg5* and *Klf4* were measured. Stem cells from low (around 5), mid-range (around 15) and high (40+) passages were used. Graphs represent mean ± SD of three independent replicates. Differences between groups were considered to be statistically significant when values of $P < 0.05$ (*).

### 3.3. DUSP7 is required for cardiomyocyte formation

Since we observed differences in the abilities of cells to form mesoderm and ectoderm at early stages, we differentiated cells in vitro for a further 5–10 days and then studied the formation of cardiac and neural cells. DUSP7 KO cells exhibited lower levels of expression of cardiomyocyte-specific transcripts (*Nkx2.5*, *Myh6*, *Myh7*) and higher levels of the expression of neuro-specific markers (*Tubb3* and *Mash1*) (Fig 4A). Similar difference in cardiomyogenesis and neurogenesis were observed also at the protein level, where DUSP7 KO cells exhibited lower levels of cardiomyocyte-specific (MHC) and higher levels of neuro-specific (βIIItubulin) proteins compared to WT (Fig 4B). To further explain the observed decreases in the expressions of cardiomyocyte specific transcripts and proteins, we studied the number of formed cardiomyocytes. Cells were cultivated for the first five days as hanging drops in order to form single EBs. Then, each EB was individually cultivated for a total of 20 days and either the whole embryoid body was stained for cardiomyocyte-specific (MHC) or neuro-specific (βIIItubulin) markers (S1 and S2 Figs) or cells were re-seeded onto a fresh plate as single cells. In the latter case, after a further day of cultivation, they were stained with antibody specific for cardiac myosin heavy chains (MF20) and with DAPI. The ratio between the number of nucleuses and myosin positive cells, which determines the number of cardiomyocytes, was lower in DUSP7 KO cell lines compared to wild type cells (Fig 5A and 5B, S3 Fig). In addition, we determined the relative number of formed cardiomyocytes on day 20 after cardiomyocytes specific selection on HG8 cells and their DUSP7 KO cells (see Material and methods). Here, we again observed that DUSP7 KO cells formed a lower number of cardiomyocytes compared to WT cells (Fig 5C). These results therefore indicate that DUSP7 is required for the formation of mouse cardiomyocytes in vitro.

### 3.4. DUSP7 depletion does not change the phosphorylation of ERK

Since DUSP7 is known to dephosphorylate MAPK with a preference towards ERK1/2, we tested whether there were differences between the levels of phosphorylated ERK, JNK and p38 at the basal level in ES cells. However, only ERK1/2 showed any differences and these were very small and deemed to be statistically insignificant (Fig 6A). Interestingly, when using siRNA for DUSP7, we also observed no change in the phosphorylation of ERK after 24h, but when using siRNA for DUSP6, we saw a stronger signal for pERK1/2. (Fig 5B). Since the phosphorylation and dephosphorylation of ERK is an important process for signal transduction and very dynamic process, we tested whether there would be a change in the kinetics of phosphorylation between wild type and KO cells. Cells were starved for 6h in media without serum. After this time, serum was added to a final concentration of 30% and phosphorylation was measured by western blot method 10min, 30min, 1h and 3h after stimulation. The highest phosphorylation was observed 10min after stimulation in all cell lines and after 1h the level of phosphorylation had returned to its basal level. Although there were slight differences between individual sets in this dynamic, neither the overall maximum level of ERK phosphorylation

(A)

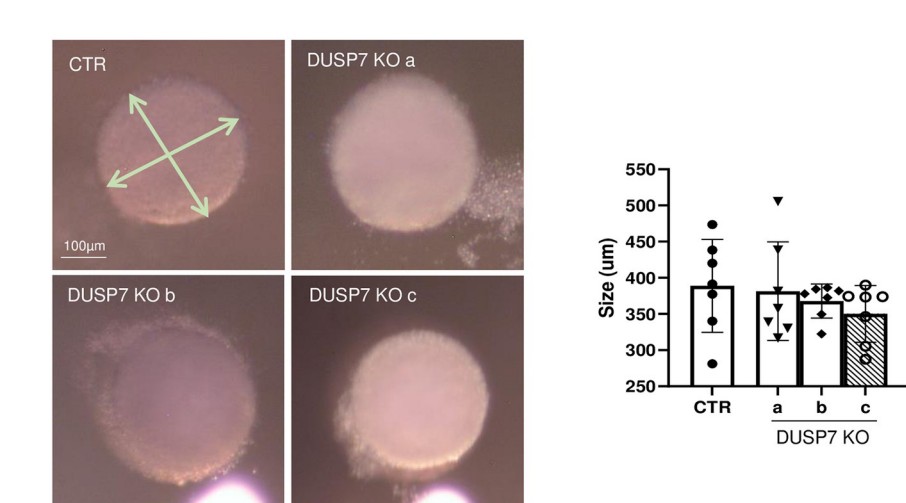

(B)

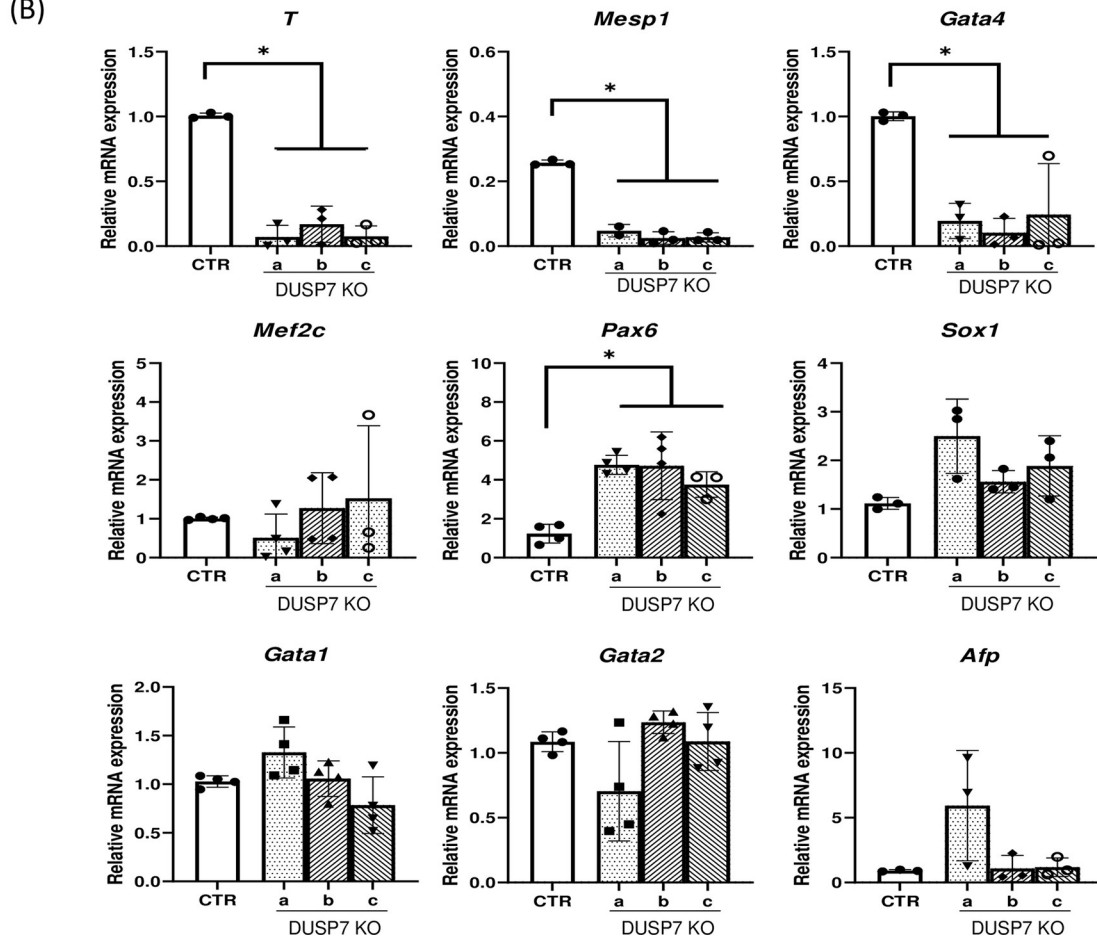

**Fig 3. DUSP7 KO cell lines are able to differentiate into all three germ layers, but preferentially express markers for ectoderm over those for mesoderm. (A)** Measures of diameter of embryoid bodies. 400 cells were used to create embryoid bodies using the hanging drop method. The sizes of embryoid bodies were measured on Day 5. Graph represents mean ± SD of seven independent replicates, each of the values representing the average value of at least 5 different measurements. **(B)** KO cell lines exhibit lower expressions of markers typical for mesoderm or cardiacmesoderm (*T*, *Mesp1*, *Gata4*) and higher expressions of markers for

ectoderm (*Sox1*, *Pax6*), while markers that characterize both myogenesis and neurogenesis (*Mef2c*) as well as endoderm markers (*Afp*) and markers for hematopoietic mesoderm (*Gata1*, *Gata2*,) have similar expression profiles as in wild type cells. Markers were measured after 5 days of differentiation. Graphs represent mean ± SD of at least three independent replicates. Differences between groups were considered to be statistically significant when values of P < 0.05 (*).

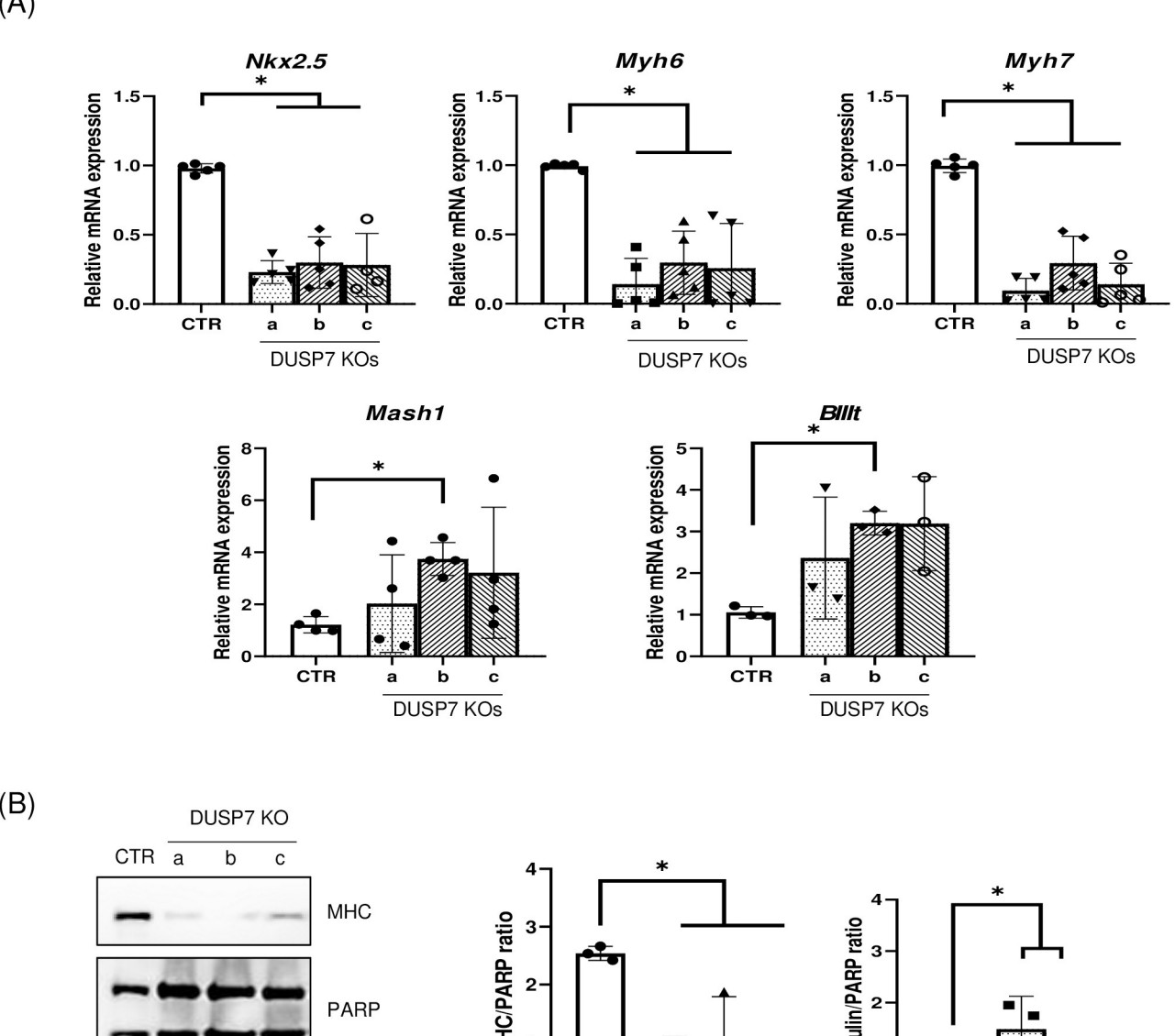

**Fig 4. DUSP7 KOs after longer differentiation in vitro express lower levels of cardiac markers compared to wild type cells. (A)** Analysis of cardiac and neural markers on qPCR. ES cells were cultivated for 10 days (Mash1) or 14 days (*Nkx2.5*, *Myh6*, *Myh7*, *BIIIt*) and analyzed on RT-qPCR, normalized to the mean expression of *Hprt* and *Rpl* (TaqMan) or *Actb* and *Tbp* (SyberGreen). Graphs represent mean ± SD of at least four independent replicates. **(B)** Western blot analysis of cardiac (MHC) and neural (BIIItubulin) markers and their quantification (on the right) after 20 days of cell differentiation. Graphs represent mean ± SD of four independent replicates. Differences between groups were considered to be statistically significant when values of P < 0.05 (*).

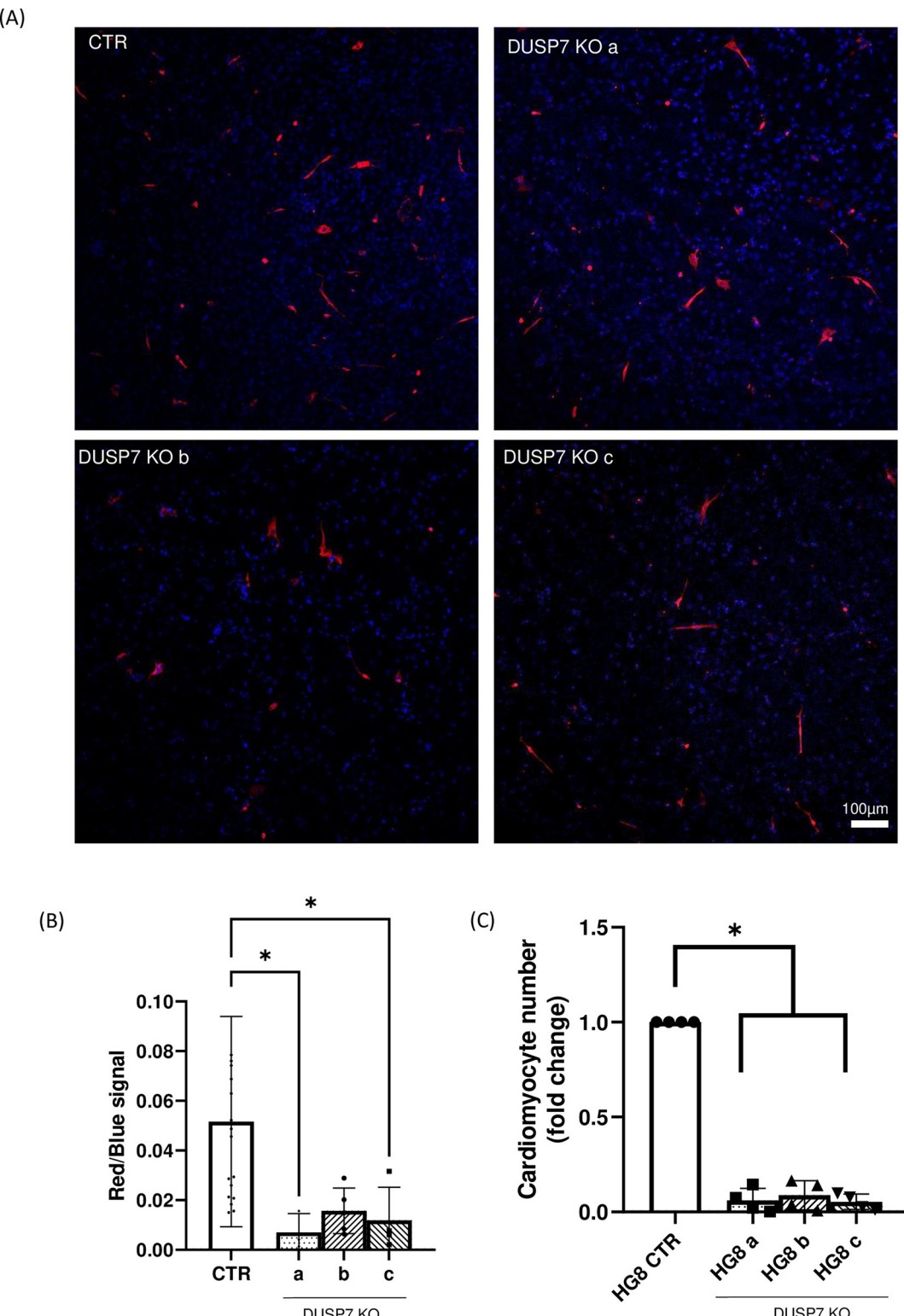

**Fig 5. DUSP7 KO cells produce a lower number of cardiomyocytes. (A)** DUSP7 KO cells form fewer cardiomyocytes compared to wild type cells, as analyzed by immunocytochemistry. Cells were stained after a total of 21 days of differentiation. Nucleus is shown in blue (DAPI) and cardiomyocytes in red, stained by cardiac specific antibody MF20. Scale bar represents 100μm. **(B)** Quantification of number of cardiomyocytes. Graph represents mean ± SD of three independent replicates, each value representing the average value for three embryoid bodies analyzed. **(C)** Number of cardiomyocytes determined in HG8

cells and their DUSP7 KO. Cell selection started on day 14 and measurements were performed on day 20. Graph represents mean ± SD of four independent replicates. Differences between groups were considered to be statistically significant when values of P < 0.05 (*).

nor the speed of dephosphorylation between wild type and KO cell lines showed statistically significant differences (Fig 6C).

### 3.5. Level of DUSP7 increases during differentiation

Since the depletion of DUSP7 did not have an effect on the basic characteristics of ES cells, (Figs 2 and 6), but did have an effect on the differentiation of cells in later stages of in vitro cultivation and on the differentiation of cardiomyocytes, we measured changes in the expression of *Dusp7* using RT-qPCR in cells from in vitro culture (Fig 7A) and in hearts of mice from different stages of development (Fig 7B). We found that the level of *Dusp7* increased over time during differentiation in culture as well as in the hearts of mice during their ontogenesis. Therefore, since DUSP7 might have a more important role in later stages of differentiation in vitro than in ES cells, we tested the level of phosphorylation of ERK1/2 in 5-day-old embryoid bodies but were not able to see any significant difference between the tested cell lines (Fig 7C).

## 4. Discussion

It is generally agreed that DUSPs specifically dephosphorylate MAPKs. However, when it comes to their specificity to individual proteins there are some conflicting reports about which substrates they can dephosphorylate, especially when it comes to the more studied phosphatases such as DUSP1 [43–47], these conflicts appearing to arise because these proteins are studied in different conditions or in different models. DUSP7 is generally believed to dephosphorylate ERK1/2, but in some conditions was shown to interact with JNK [48–52] However, there are also studies which suggest the possibilities of DUSP7 dephosphorylating substrates other than members of the MAPK family. It has been shown that DUSP7 can also dephosphorylate cPKC isoforms [53], thus inhibiting their activity. In the case of DUSP7 depletion, the activity of cPKC is not inhibited at the correct time or for the correct duration, which leads to defects in meiosis in mouse oocytes; however, our data suggest that the depletion of DUSP7 does not affect the mitosis of ES cells (Fig 2B).

There are numerous studies which show the effects of the activation of MAPK/ERK in ES cells on their stemness and differentiation [54–56]. The depletion of DUSP2 and its effect on ES cells was also studied in association with DUSP7 by Chappell et al., who showed that DUSP7 is necessary for the preservation of pluripotency [57]. However, a significant effect on pluripotency was shown only when DUSP7 was overexpressed, or when DUSP7 was knocked-down together with DUSP2. In contrast, our model shows that ES cells are able to adapt to long cultivation when DUSP7 is knocked-out by itself (Fig 2D).

During differentiation in KO cells, we observed a significant decrease in the general mesodermal marker *T* as well as in *Mesp1*, which appears in the cardiogenic area of the primitive [58]. The expression of *Gata4*, a gene necessary for normal heart tube formation [40, 59] and a regulator of other genes critical for cardiomyogenesis [60], was also downregulated, unlike its cofactor *Mef2c*, which is also expressed during the early development of myocardium and other muscle cells [39, 61, 62]. The *Mef2c* marker was more variable between KO lines, but did not show a significant decrease or increase compared to control. Although this gene is widely used as a cardiac marker, it is greatly expressed also in mouse brain [63] and is crucial for normal neural development [64]; therefore, its potentially lower levels due to reduced cardio myogenesis are masked by potentially higher levels in developing neural progenitors. Preferential

(A)

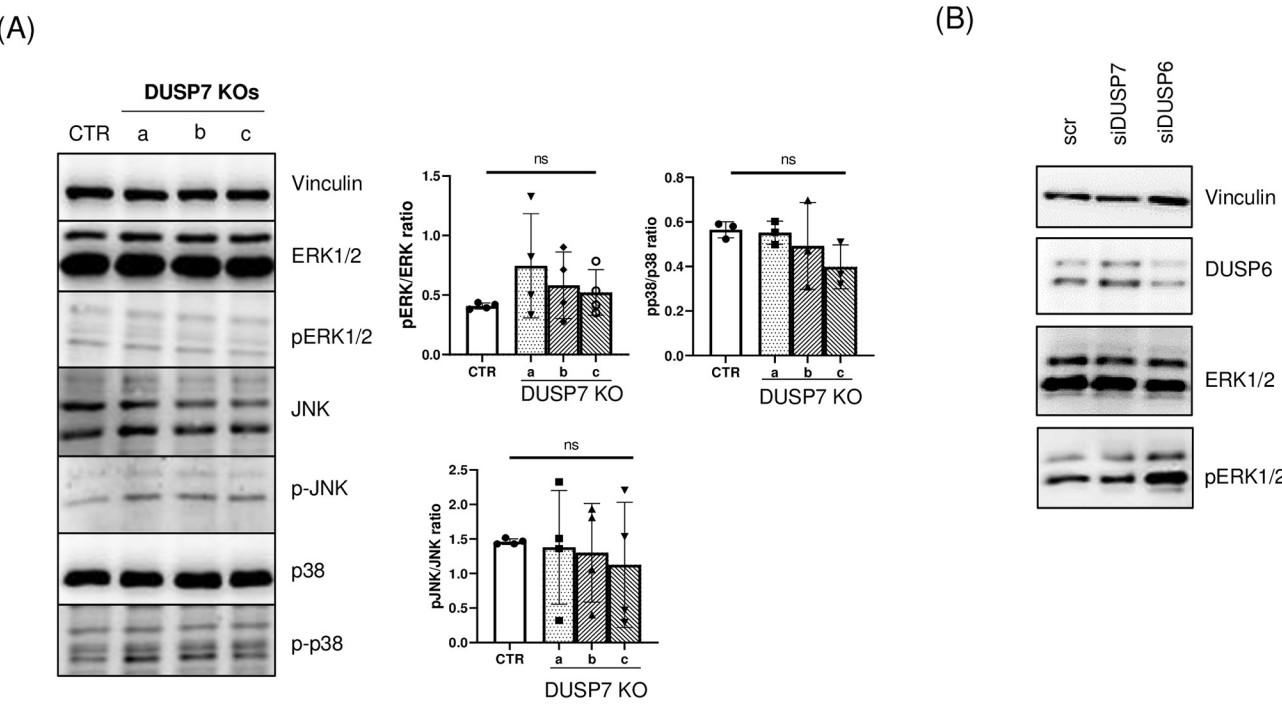

(B)

(C)

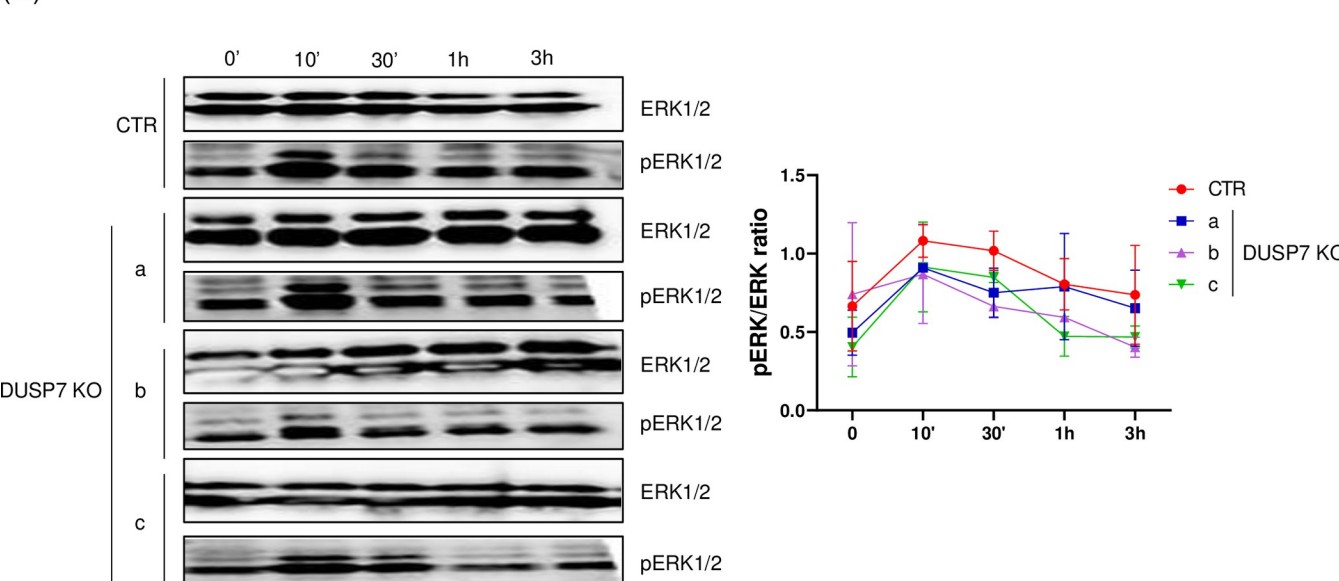

**Fig 6. Level of phosphorylation of MAPK is the same in DUSP7 KO cells.** (A) Levels of phosphorylation of different MAPKs were measured in unstimulated wild type and DUSP7 KO ES cells. Quantitave analysis of the ratios between total ERK1/2 and pERK1/2, JNK and pJNK, and p38 and pp38 are shown (right). Graphs represent mean ± SD of four independent replicates. (B) Downregulation of DUSP7 by siRNA has no effect on the phosphorylation of ERK1/2. Cells were transfected by siRNA 24h after passage and lysed after a further 24h of cultivation. Transfection by scrambled siRNA (scr) was used as control. (C) ES cells were cultivated in serum free medium for 6h (time point 0') and then the phosphorylation of ERK1/2 was stimulated by adding FBS to a total 30% concentration for 10min, 30min, 1h and 3h before analysis. Quantitative analysis of the ratio between total ERK1/2 and pERK1/2 is shown (right). Graph represents mean ± SD of two independent replicates.

(A)

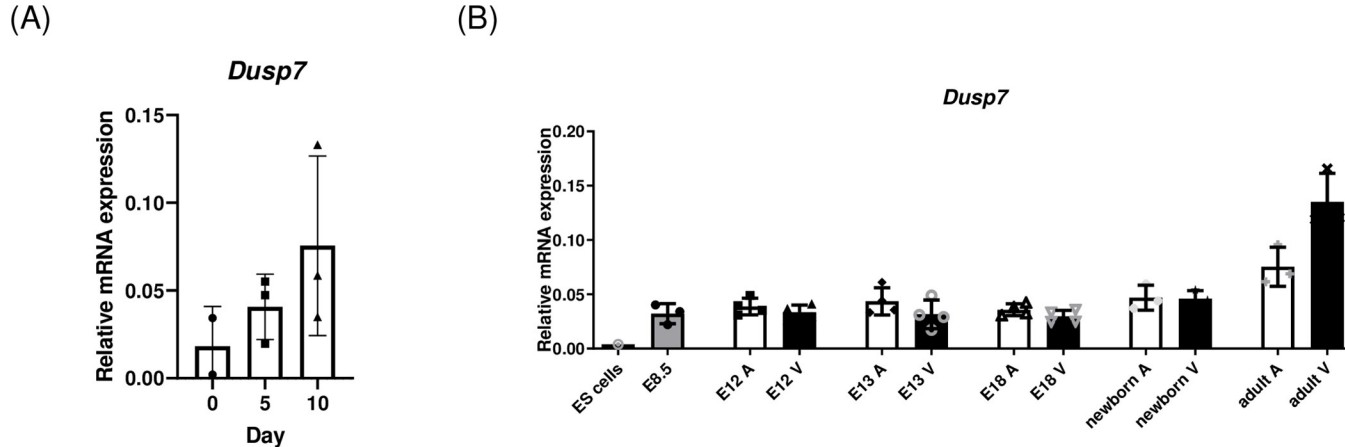

(B)

(C)

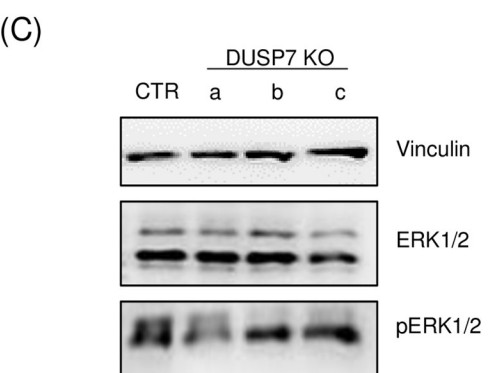

**Fig 7. Level of DUSP7 changes over time during differentiation.** (A) Changes in the level of *Dusp7* expression in vitro culture analyzed by qPCR. Graph represents mean ± SD of three independent replicates. (B) Level of *Dusp7* in ES cells and murine heart at different stages of its development analyzed by qPCR. The level of *Dusp7* has the same pattern in both atrium (A) and ventriculus (V) at each individual time-point analyzed, but it changes over time, with its lowest expression seen in ES cells and its highest in adult hearts. Graph represents mean ± SD of at least three independent replicates. Differences between groups were considered to be statistically significant when values of P < 0.05 (*). (C) Phosphorylation of ERK1/2 in embryoid bodies after 5d of in vitro cultivation.

differentiation of DUSP7 KO into neuro-ectodermal linages is supported by elevated levels of *Sox1* and *Pax6*. In contrast to cardiac-mesoderm markers, the levels of mesodermal markers *Gata1* and *Gata2*, which are important for the formation of hematopoietic lineages, were not changed by the depletion of DUSP7. The importance of different MAPK in hematopoiesis and especially in diseases such as leukemia has been studied, but it has been shown that p38α plays the key role in hematopoietic stem cell activation and, later, in their maturation during hematopoiesis [65]. With respect to the interaction of DUSP7 with members of the MAPK family, there is least evidence that DUSP7 interacts with p38; therefore, the fact of its depletion having no effect on hematopoietic markers was to be expected.

Several members of the DUSP family have been studied with respect to the development of heart tissue or the neural system, most of them via the mechanism linked with the dephosphorylation of ERK. When it comes to neural development, DUSP1, DUSP4 and DUSP6 were shown to be regulated by nerve growth factor [11, 66, 67]. DUSP1 is necessary for normal axonal branching [68], similarly to DUSP6, which also plays a role in normal axon development [69]. The overexpression of DUSP1 has a neuroprotective effect in response to ischemia [70,

71] and, together with DUSP4, they protect motor axons from degradation [72]. The role of DUSP7 in neural development has not yet been thoroughly studied, despite DUSP7 being expressed in whole brain of mice [73]. Our study, therefore, is one of the first to show that DUSP7 can inhibit the development of neuronal lineages in an in vitro model of ES cell differentiation.

When it comes to the development of heart, MAPK play an important role since they are highly involved in FGF and BMP signaling–two very important signaling pathways playing a role in cardiac mesoderm and myocardium formation. These pathways need to be almost periodically activated and inhibited for normal formation of heart, which can be achieved by negative feedback mediated by MAPK-induced DUSP expression [2, 74].

In heart, DUSPs have mostly been linked with the regulation of the ERK signaling pathway and the proliferation of cardiomyocytes. In general, all studies involving the depletion of any DUSPs show heart enlargement. However, in some cases, there need to be special conditions like heart exposure to hypoxia for hypertrophy to be apparent [75], or hypertrophy is detectable only in adults or after injury, such as in DUSP6-deficient fish [76]. Depletions of different DUSPs in mice have shown changes in cardiomyocyte morphology (DUSP8, [77]), or have been linked to protection (DUSP6, [78]) or, in contrast, to increased susceptibility to cardiomyopathy (DUSP1, DUSP4, [79]). All of these studies either operate with the measurement of hearts of adult subjects or, in the case of in vitro studies, they use already differentiated or neonatal cardiomyocytes, in contrast to our study, which investigated differentiation from ES cells and the effect of DUSP7 on early cardiomyocyte differentiation.

As mentioned before, an appropriate level of activation at the right time and for the right duration plays an important role in the development of heart. For example, the activation of ERK by 12-O-Tetradecanoylphorbol-13-acetate (TPA) leads to an increase in cardiomyogenesis, but only when the treatment is applied in a certain time window [22]. Similarly, we see in our experiments that applying TPA at the indicated time does increase the number of cardiomyocytes in our KO cell lines (S4 Fig) comparably to wild types; however, already at this point, there are fewer mesodermal cells on which it can have an impact, as we showed by measuring *T* and *Mesp1* expression. Furthermore, cardiomyocytes derived from KO cells have the same maturation profile as WT, as shown by the ratios of *Nkx2.5*, *Myh6*, and *Myh7* [22] (S5 Fig). Therefore, it seems that DUSP7 plays a role in early stages of this process and does not have a big impact on later cardiomyocyte development.

Since, as mentioned, DUSP7 is specific towards MAPK with a preference towards ERK1/2 [15], we studied the levels of phosphorylation of different MAPK, but, we did not observe any significant differences in our KO cell lines compared to wild type. This is in contrast to previously published observations; however, some of these publications show only the slightly higher phosphorylation of ERK1/2 when the expression of DUSP7 is lowered in combination with other DUSPs [57] or under special conditions, such as in DUSP7 KO mice that are on a high fat diet [80]. The observation of only a very small effect of DUSP7 could also be due to the fact that DUSP7 exhibited very low expression in our ES cells to begin with, more than 10x lower compared, for example, to DUSP6 (S6 Fig), which is reinforced by the fact that when using siRNA for DUSP6 and DUSP7 we could see changes in the phosphorylation of ERK only in the former case (Fig 5B). However, unlike in many studies which used siRNA or shRNA, or measured the phosphorylation levels a short time after adding some inhibitors or activators, our cells were modified using CRISPR/Cas9 and were cultivated for a long time, during which they might have adapted to DUSP depletion. This can be also demonstrated by DUSP6, where we did not see the same effect when it was knocked out using CRISPR/Cas9 as when using siRNA (S7 Fig). We also saw that the level of DUSP7 mRNA rises during differentiation in vitro and in developing heart, indicating its importance in such development, including the

growth of myocardium. Since in our experimental design cells were being differentiated through the formation of EBs, and culture conditions throughout the differentiation process did not favor any individual cell type in particular, we can assume that the ones which achieved a head start overgrew in the culture and "smothered" other cell types, whose differentiation might have been compromised by genetic modification and which would have appeared later in the culture.

## 5. Conclusions

In summary, on the basis of all of our results, we can conclude that DUSP7 promotes early differentiation towards neural cells and that in DUSP7 KO early cardiac mesoderm is repressed, which, in prolonged cultivation, is reflected by a lower number of formed cardiomyocytes.

## Supporting information

**S1 Fig.** A: Whole embryoid body staining of WT cells, B: Whole embryoid body staining of DUSP7 KOa cells, C: Whole embryoid body staining of DUSP7 KOb cells, D: Whole embryoid body staining of DUSP7 KOc cells.
(TIF)

**S2 Fig.** A: Whole embryoid body staining of WT cells, B: Whole embryoid body staining of DUSP7 KOa cells, C: Whole embryoid body staining of DUSP7 KOb cells, D: Whole embryoid body staining of DUSP7 KOc cells.
(TIF)

**S3 Fig.** A: DUSP7 KO cells form fewer cardiomyocytes compared to wild type cells, as analyzed by immunocytochemistry–WT cells, B: DUSP7 KO cells form fewer cardiomyocytes compared to wild type cells, as analyzed by immunocytochemistry–DUSP7 KOa cells, C: DUSP7 KO cells form fewer cardiomyocytes compared to wild type cells, as analyzed by immunocytochemistry–DUSP7 KOb cells, D: DUSP7 KO cells form fewer cardiomyocytes compared to wild type cells, as analyzed by immunocytochemistry–DUSP7 KOc cells.
(TIF)

**S4 Fig. Addition of TPA increases the number of cardiomyocytes.**
(TIF)

**S5 Fig. Cardiomyocytes derived from DUSP7 KO cells have the same maturity profile as cardiomyocytes from WT cells.**
(TIF)

**S6 Fig. Expression of DUSP changes during development.**
(TIF)

**S7 Fig. Difference in the phosphorylation of ERK1/2 by the downregulation of DUSP6.**
(TIF)

**S1 Raw images.**
(PDF)

## Author Contributions

**Conceptualization:** Stanislava Sladeček, Katarzyna Anna Radaszkiewicz, Jiří Pacherník.

**Funding acquisition:** Jiří Pacherník.

**Investigation:** Stanislava Sladeček, Katarzyna Anna Radaszkiewicz.

**Methodology:** Stanislava Sladeček, Katarzyna Anna Radaszkiewicz, Martina Bőhmová, Tomáš Gybeľ, Tomasz Witold Radaszkiewicz.

**Writing – original draft:** Stanislava Sladeček.

**Writing – review & editing:** Katarzyna Anna Radaszkiewicz, Jiří Pacherník.

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
