## [Decision Letter · Decision Letter 0]

7 Jun 2022

PONE-D-22-10089Dual Specificity Phosphatase 7 Drives the Formation of Cardiac Mesoderm in Mouse Embryonic Stem CellsPLOS ONE

Dear Dr. Pachernik,

Thank you for submitting your manuscript to PLOS ONE. After careful consideration, we feel that it has merit but does not fully meet PLOS ONE’s publication criteria as it currently stands. Therefore, we invite you to submit a revised version of the manuscript that addresses the points raised during the review process.

Please revise your manuscript by addressing all the comments raised by the reviewers that are given below.

We look forward to receiving your revised manuscript.

Kind regards,

Johnson Rajasingh, Ph.D, HCLD

Academic Editor

PLOS ONE

Journal Requirements:

2.We note that the grant information you provided in the ‘Funding Information’ and ‘Financial Disclosure’ sections do not match. 

"This research was supported by the Faculty of Science of Masaryk University (MUNI/A/1145/2017) and by the Czech Science Foundation (Project 18-18235S); SS was supported by a grant from the Czech Science Foundation (Project 19-16861S)."

"This research was supported by the Faculty of Science of Masaryk University (MUNI/A/1145/2017) and by the Czech Science Foundation (Project 18-18235S); SS was supported by a grant from the Czech Science Foundation (Project 19-16861S). 

https://www.muni.cz/en/about-us/organizational-structure/faculty-of-science

https://gacr.cz/en/

4.PLOS ONE now requires that authors provide the original uncropped and unadjusted images underlying all blot or gel results reported in a submission’s figures or Supporting Information files. This policy and the journal’s other requirements for blot/gel reporting and figure preparation are described in detail at https://journals.plos.org/plosone/s/figures#loc-blot-and-gel-reporting-requirements and https://journals.plos.org/plosone/s/figures#loc-preparing-figures-from-image-files. When you submit your revised manuscript, please ensure that your figures adhere fully to these guidelines and provide the original underlying images for all blot or gel data reported in your submission. See the following link for instructions on providing the original image data: https://journals.plos.org/plosone/s/figures#loc-original-images-for-blots-and-gels. 

Reviewers' comments:

Reviewer's Responses to Questions

**Comments to the Author**

1. Is the manuscript technically sound, and do the data support the conclusions?

Reviewer #1: Yes

Reviewer #2: Yes

2. Has the statistical analysis been performed appropriately and rigorously? 

Reviewer #1: Yes

Reviewer #2: Yes

3. Have the authors made all data underlying the findings in their manuscript fully available?

Reviewer #1: Yes

Reviewer #2: Yes

4. Is the manuscript presented in an intelligible fashion and written in standard English?

Reviewer #1: Yes

Reviewer #2: Yes

5. Review Comments to the Author

Reviewer #1: In this article, the authors analyzed the role of DUSP7 in cardiac differentiation using mouse embryonic stem cells. They utilized CRISPR technology to generate DUSP7 KO cells and supplemented their results with qPCR and Western Blot analysis. Overall, I recommend this paper be accepted with minor revisions because I believe the abstract and some of the figures need improvement. Additionally, I have a couple of questions/concerns about their cell studies.

The abstract should focus on the work done by the authors only and should not include any references. The cells were cultured for more than 40 passages, did they perform any karyotyping to determine if any chromosomal abnormalities were present in the cells after long shelf life? Why not perform overexpression of DUSP7 if you are talking about it in your discussion? I believe overexpression studies would have given strong support to your paper.

In Figures 1B and 1D, for example, the font is too small and some of the labels are not legible. The quality of Figure 3 is very poor, either improve the resolution or substitute for the supplemental immunostaining pictures. Figure 5 needs to show a cardiomyocyte-specific marker/antibody.

Reviewer #2: Sladecek et al. research article titled “Dual specificity phosphatase 7 drives the formation of cardiac mesoderm in mouse embryonic stem cells” focuses on the role of DUSP7 in mouse embryonic stem cells and elucidates that the DUSP7 is more important for early neural and cardiac mesoderm development by in vitro. The data presented in this study strongly support the conclusion drawn by the authors.

Some of the minor concerns are noted below to improve this manuscript before publishing.

1. Typo errors need to be checked.

a. In the abstract, there are some numerical numbers in each line (21 -31). It doesn’t make any meaning.

b. In section 3.5, DUSP7 is mentioned in small/capital letters. Please make it uniform throughout the manuscript.

2. In fig. 5A, the scale bar was not visible clearly.

3. The resolution of the figures is not good. Importantly the figure.1 text is hard to read. Please provide high quality figures.

6. PLOS authors have the option to publish the peer review history of their article (what does this mean?). If published, this will include your full peer review and any attached files.

Reviewer #1: No

Reviewer #2: No

---

## [Author Response · Author response to Decision Letter 0]

29 Jun 2022

Dear editor and reviewers,

Thank you very much for your comments to our manuscript. Please find bellow our responses to individual raised points. We hope they will be to your satisfaction.

Reviewer #1:

The abstract should focus on the work done by the authors only and should not include any references.

References, numbers from the abstract, have been removed.

The cells were cultured for more than 40 passages, did they perform any karyotyping to determine if any chromosomal abnormalities were present in the cells after long shelf life?

All experiments in which cells were differentiated as well as experiments based on which we were drawing conclusions on DUSP7’s effect on phosphorylation have been done in cells cultured 5-20 passages. The high passage was only used in experiment involving markers of pluripotency (Fig. 2D) to see if there will be some change in our KO cell lines in pluripotency even after longer shelf life. However, since we did not see any difference between the KO and WT cells, we did not investigate further changes that this kind of long-term cultivation could have. We have not performer karyotyping of the cells ourselves, but the used mouse embryonic stem cells line R1 (from which also all the KO cells lines were derived) is well established and defined cell line, which we have bought from ATCC (American Tissue Culture Collection).

Why not perform overexpression of DUSP7 if you are talking about it in your discussion? I believe overexpression studies would have given strong support to your paper.

Currently we are still working on determining the importance of DUSP7 as well as couple of other DUSPs in the development of early mesoderm and cardiomyocytes, where our preliminary results point to their importance in different stages of development. In that context, we agree that it would be very interesting to investigate the overexpression of DUSP7. However, this would be more interesting when it comes to pinpointing the importance of DUSPs in different stages, developmental intervals, than overall general effect that DUSP7 has in in vitro development from mouse ES cells, which we report in this paper.

In Figures 1B and 1D, for example, the font is too small and some of the labels are not legible.

We have exported pictures in high resolution and hope that this solves the issue of the labels not being legible.

The quality of Figure 3 is very poor, either improve the resolution or substitute for the supplemental immunostaining pictures.

We have exported the figures in high resolution (330dpi) which we hope resolves the issue. The pictures of embryoid bodies in Figure 3 were taken in hanging drops on stereomicroscope which does not provide very good quality of magnification. Furthermore, embryoid bodies are floating 3D objects that are difficult to focus on. Their pictures were used here more as an illustration to the graph next to them which shows the size of embryoid bodies on day 5. We cannot substitute them with immunostaining pictures from supplements, since those are taken on day 20 and have been cultured as adherent cells for 15 days. We hope that the quality in which they are now exported is acceptable.

Figure 5 needs to show a cardiomyocyte-specific marker/antibody.

We have changed the description of this figure, which now states that MF20 was used antibody instead of MHC. As stated in the material and methods (table 2) we used clone MF20 from the Developmental Studies Hybridoma Bank, which is specific for myosin heavy chain 6 and 7, which are considered as cardiomyocyte-specific markers.

Reviewer #2:

Typo errors need to be checked.

a. In the abstract, there are some numerical numbers in each line (21 -31). It doesn’t make any meaning.

This has been addressed and the numbers have been erased

b. In section 3.5, DUSP7 is mentioned in small/capital letters. Please make it uniform throughout the manuscript.

We call our cell lines DUSP7 KOa/b/c and generally when speaking about DUSP7, to have it more uniformed, we treat it as speaking about the mouse protein rather than gene and therefore write it all in capital letters. However, in case of section 3.5 as well as when its mentioned in Table 1 we are explicitly talking about measuring the gene expression of DUSP7 and therefore we used the the nomenclature which is used for mouse genes – italicized, with only the first letter in upper-case. We had mistaken in this regard in section 2.5 and Figure 1 which has been corrected.

In fig. 5A, the scale bar was not visible clearly.

We modified scale bar in fig. 5A and added it (Scale bar represents 100um) into the description of fig.5.

The resolution of the figures is not good. Importantly the figure.1 text is hard to read. Please provide high quality figures.

We have exported figures in high quality (330 dpi) which we hope resolves the issue.

---

## [Decision Letter · Decision Letter 1]

26 Sep 2022

Dual Specificity Phosphatase 7 Drives the Formation of Cardiac Mesoderm in Mouse Embryonic Stem Cells

PONE-D-22-10089R1

Dear Dr. Pachernik,

We’re pleased to inform you that your manuscript has been judged scientifically suitable for publication and will be formally accepted for publication once it meets all outstanding technical requirements.

Kind regards,

Johnson Rajasingh, Ph.D, HCLD

Academic Editor

PLOS ONE

Additional Editor Comments (optional):

Reviewers' comments:

Reviewer's Responses to Questions

**Comments to the Author**

1. If the authors have adequately addressed your comments raised in a previous round of review and you feel that this manuscript is now acceptable for publication, you may indicate that here to bypass the “Comments to the Author” section, enter your conflict of interest statement in the “Confidential to Editor” section, and submit your "Accept" recommendation.

Reviewer #1: All comments have been addressed

Reviewer #2: All comments have been addressed

2. Is the manuscript technically sound, and do the data support the conclusions?

Reviewer #1: Yes

Reviewer #2: Yes

3. Has the statistical analysis been performed appropriately and rigorously? 

Reviewer #1: Yes

Reviewer #2: Yes

4. Have the authors made all data underlying the findings in their manuscript fully available?

Reviewer #1: Yes

Reviewer #2: Yes

5. Is the manuscript presented in an intelligible fashion and written in standard English?

Reviewer #1: Yes

Reviewer #2: Yes

6. Review Comments to the Author

Reviewer #1: The authors addressed all the reviewer's comments and corrected the figures specified in the first revision.

Reviewer #2: The Authors have rectified and addressed all the pointed comments in the revised version of the manuscript.

7. PLOS authors have the option to publish the peer review history of their article (what does this mean?). If published, this will include your full peer review and any attached files.

Reviewer #1: No

Reviewer #2: No

---

## [Editor Report · Acceptance letter]

30 Sep 2022

PONE-D-22-10089R1 

Dual specificity phosphatase 7 drives the formation of cardiac mesoderm in mouse embryonic stem cells 

Dear Dr. Pacherník:

I'm pleased to inform you that your manuscript has been deemed suitable for publication in PLOS ONE. Congratulations! Your manuscript is now with our production department. 

Kind regards, 

on behalf of

Dr. Johnson Rajasingh 

Academic Editor

PLOS ONE